# Nutritional Supplementation Reduces Lesion Size and Neuroinflammation in a Sex-Dependent Manner in a Mouse Model of Perinatal Hypoxic-Ischemic Brain Injury

**DOI:** 10.3390/nu14010176

**Published:** 2021-12-30

**Authors:** Myrna J. V. Brandt, Cora H. Nijboer, Isabell Nessel, Tatenda R. Mutshiya, Adina T. Michael-Titus, Danielle S. Counotte, Lidewij Schipper, Niek E. van der Aa, Manon J. N. L. Benders, Caroline G. M. de Theije

**Affiliations:** 1Department for Developmental Origins of Disease, University Medical Center Utrecht Brain Center and Wilhelmina Children’s Hospital, Utrecht University, 3508 AB Utrecht, The Netherlands; M.J.V.Brandt-3@umcutrecht.nl (M.J.V.B.); C.Nijboer@umcutrecht.nl (C.H.N.); 2Centre for Neuroscience, Surgery and Trauma, Blizard Institute, Queen Mary University of London, London E1 2AD, UK; i.nessel@qmul.ac.uk (I.N.); t.r.mutshiya@qmul.ac.uk (T.R.M.); a.t.michael-titus@qmul.ac.uk (A.T.M.-T.); 3Danone Nutricia Research, 3508 TC Utrecht, The Netherlands; Danielle.Counotte@danone.com (D.S.C.); Lidewij.Schipper@danone.com (L.S.); 4Department of Neonatology, University Medical Center Utrecht Brain Center and Wilhelmina Children’s Hospital, Utrecht University, 3508 AB Utrecht, The Netherlands; N.vanderAa@umcutrecht.nl (N.E.v.d.A.); M.Benders@umcutrecht.nl (M.J.N.L.B.)

**Keywords:** hypoxic-ischemic encephalopathy, neonate, fish oil, DHA, EPA, UMP, choline, iodide, zinc, vitamin B12, neuroinflammation, neurodevelopment, diet, mouse

## Abstract

Perinatal hypoxia-ischemia (HI) is a major cause of neonatal brain injury, leading to long-term neurological impairments. Medical nutrition can be rapidly implemented in the clinic, making it a viable intervention to improve neurodevelopment after injury. The omega-3 (*n*-3) fatty acids docosahexaenoic acid (DHA, 22:6*n*-3) and eicosapentaenoic acid (EPA, 20:5*n*-3), uridine monophosphate (UMP) and choline have previously been shown in rodents to synergistically enhance brain phospholipids, synaptic components and cognitive performance. The objective of this study was to test the efficacy of an experimental diet containing DHA, EPA, UMP, choline, iodide, zinc, and vitamin B12 in a mouse model of perinatal HI. Male and female C57Bl/6 mice received the experimental diet or an isocaloric control diet from birth. Hypoxic ischemic encephalopathy was induced on postnatal day 9 by ligation of the right common carotid artery and systemic hypoxia. To assess the effects of the experimental diet on long-term motor and cognitive outcome, mice were subjected to a behavioral test battery. Lesion size, neuroinflammation, brain fatty acids and phospholipids were analyzed at 15 weeks after HI. The experimental diet reduced lesion size and neuroinflammation specifically in males. In both sexes, brain *n*-3 fatty acids were increased after receiving the experimental diet. The experimental diet also improved novel object recognition, but no significant effects on motor performance were observed. Current data indicates that early life nutritional supplementation with a combination of DHA, EPA, UMP, choline, iodide, zinc, and vitamin B12 may provide neuroprotection after perinatal HI.

## 1. Introduction

Perinatal hypoxia-ischemia (HI) is a leading cause of neonatal brain injury [1]. Of infants surviving HI, ~30% suffer from long-term motor and neurological impairments [1,2]. Principal among these sequelae are cerebral palsy, epilepsy and mental retardation in more severe cases, while children with a milder form of HI encephalopathy (HIE) can show impairments across several cognitive domains [2]. HIE in infants born at term can result in severe grey matter injury, affecting various cortical and deep brain structures, such as the sensorimotor cortex, hippocampus, thalamus and basal ganglia [3]. Boys are most susceptible to injury, with worse outcome than girls, possibly due to larger fetal size, hormonal and/or genetic differences [1], or enhanced immune activation [4,5].

The only effective treatment currently available for HIE is hypothermia for 72 h, starting within 6 h after birth [6,7]. However, since hypothermia only provides partial protection [7,8], novel therapies to further improve outcome after HI are urgently needed. Medical nutrition is a viable option for neuroprotection and neurorepair, as it can be rapidly implemented in the clinic and is generally considered safe [9].

Phospholipids such as phosphatidylcholine (PC) and phosphatidylethanolamine (PE) are important components of neural and glial membranes, synapses, and myelin [10]. *n*-3 polyunsaturated fatty acids (PUFAs) such as docosahexaenoic acid (DHA, 22:6*n*-3) and eicosapentaenoic acid (EPA, 20:5*n*-3), in addition to uridine monophosphate (UMP) and choline, are required to synthesize PC through a physiological process known as the Kennedy cycle. Synthesis of these phospholipids can be synergistically increased by dietary supplementation of their precursors, leading to enhanced functional connectivity and cognitive performance [10,11]. Zinc and vitamin B12 are also involved in PC synthesis, by acting as a cofactor for the production of methionine needed for the methylation of PE in order to generate PC [12,13,14]. The combined supplementation of these nutrients after injury may therefore contribute to functional repair [15].

In addition to their (synergistic) effects in enhancing brain phospholipid synthesis, the *n*-3 fatty acids and micronutrients are known to positively influence brain development through other mechanisms. Perinatal supply of DHA and EPA has been shown to support neural and retinal development [16]. Furthermore, the neuro-regenerative and anti-inflammatory effects of *n*-3 fatty acids, and particularly of DHA, have been widely studied in animal models of ischemic brain injury and have been shown to reduce apoptosis and inflammation (see [17] for a review), while depletion impairs recovery in a model of traumatic brain injury [18]. Zinc, iodide and vitamin B12 are essential contributors to brain development through their role in neurogenesis, synaptic connectivity, DNA methylation and increasing the levels of growth factors, ultimately leading to enhanced cognition [14,19,20,21,22,23,24]. Furthermore, these micronutrients can stimulate neuronal survival after injury either directly or through the reduction of oxidative stress [21,22,25,26].

In a double-blind randomized-controlled trial (RCT), the effects of nutritional supplementation containing DHA, ARA, UMP, multivitamins, trace elements and minerals have been studied in term and preterm infants with perinatal brain injury [27]. Results have shown clinically relevant increases in Bayley-III cognitive and language scores after 1-2 years of dietary supplementation in infants with confirmed or suspected cerebral palsy [15] and infants at risk of neurodevelopmental impairments [28]. Larger-scale clinical trials using similar nutritional supplements are currently ongoing (NL9814, Netherlands Trial Register, The Netherlands, and NIHR130925, National Institute of Health Research, Great Britain).

In the current study, we investigated the potentially beneficial effects of an experimental diet containing DHA, EPA, UMP, choline, iodide, zinc, and vitamin B12 on lesion size, neuroinflammation and behavioral outcome in a well-validated mouse model of perinatal HIE [29]. In addition, we studied the effects of the experimental diet on total and phospholipid-bound fatty acids and brain phospholipids.

## 2. Materials and Methods

### 2.1. Animals and Diets

This study was conducted in accordance with institutional guidelines for the care and use of laboratory animals of Utrecht University and the University Medical Center Utrecht, and all animal procedures related to the purpose of the research were approved by the local Animal Welfare Body (AWB; Utrecht, The Netherlands) under an Ethical license provided by the national competent authority (*Centrale Commissie Dierproeven*, CCD, The Netherlands), securing full compliance with the European Directive 2010/63/EU for the use of animals for scientific purposes. The experimental design was subsequently approved by the AWB. All efforts were made to report animal experiments according to the ARRIVE guidelines [30].

The number of animals needed for this study was determined using G*power software (version 3.1, [31]), based on the 2 × 2 (HI-status × diet) study design, an alpha of 1.25% (5% over a family of four comparisons), power of 90% and effect size of 1.3, leading to an *n* = 20 for the primary outcome method (lesion size as determined by histology). Standard deviation was calculated at 10% from experiments previously performed by our group (Department for Developmental Origins of Disease; DDOD, UMC Utrecht, Utrecht, The Netherlands) using the same mouse model for HI injury at P9 [29].

Animals were housed under open cage (length × width × height = 267 × 208 × 140 mm) housing conditions, with woodchip bedding, plastic shelters and tissues provided, on a 12 h day/night cycle (lights on at 7:00), in a temperature-controlled room at 20–24 °C and 45–65% humidity. C57Bl/6 mice (C57Bl/6J OlaHsd, Envigo, Horst, The Netherlands), the most commonly used strain for this experimental model (see below), were bred in-house by placing immune-competent, wild type males and females together in a ratio of 1:1 or 1:2 for two weeks. Afterwards, dams were housed individually for approximately one week until delivery. Breeding mice were provided with water and rat/mouse maintenance chow (V1534-000, ssniff Spezialdiäten, Soest, Germany) *ad libitum* until pups were born. Breeding mice were used until they reached nine months of age and were not reused once they had received either of the diets used for this study (see below).

On the day of birth (postnatal day 0; P0), dams and pups were randomly assigned to either a semisynthetic experimental diet (see Table 1) or an isocaloric semisynthetic control diet (AIN93G; ssniff Spezialdiäten). Experimenters were blinded to diets throughout the study. The diets were kept at −20 °C until use, and they were replaced twice a week. Furthermore, at P0, litters were culled to a maximum of 9 pups, to ensure adequate feeding of each pup, while striving for an even male/female ratio within each litter. To induce HIE, pups were subjected to the Vanucci–Rice model, the most commonly used and well-validated model of HIE in newborn rodents. In this model, the common carotid artery is occluded unilaterally, followed by a period of systemic hypoxia [29,32]. When performed at P7–10 in mice, the model induces brain damage corresponding to that of a term HI-injured infant, with severe cortical and deep grey matter lesions and motoric and cognitive impairments (e.g., [33,34]) and more severe immune activation in males [4,5]. At P9, pups were anesthetized using isoflurane (4–5% at induction and 1–2% during maintenance), and the right common carotid artery was permanently ligated. Control animals underwent sham surgery during which they were anesthetized and the carotid artery was isolated but not ligated. 2% xylocaine/0.5% bupivacaine was applied to the wound for local anesthesia. All pups underwent toe marking for identification purposes. HI injury or sham-surgery was randomly assigned within litters, taking into account the ratio of males and females within each litter. After surgery, pups were returned to the home cage for a period of at least 75 min. Pups that underwent carotid artery ligation were then placed in a temperature-controlled, humidified hypoxic chamber (10% O_2_) for 45 min. Because of the risk of mortality during or after HI surgery, litters were additionally culled to a maximum of 7 pups on the day after surgery (see Table 2).

Mice were weaned exactly four weeks after surgery and housed with same-sex littermates or same-sex mice that were in the same diet group (Table 2). If no cage mate was available, mice were housed solitarily (*n* = 3, all male). It was later tested and confirmed that solitary housing did not affect the outcome parameters (data not shown). After weaning, mice were kept on their respective diet and received tap water *ad libitum* until the end of the experiment. Food consumption per litter (before weaning) or per cage (after weaning) was monitored throughout the experiment by weighing the food on a bi-weekly base (Figure 1A–C). Mice were sacrificed at 15 weeks after HI surgery, on P114, by an intraperitoneal injection of 0.1 mL 20% pentobarbital, followed by either perfusion or decapitation. The experiment was carried out in six partially overlapping cohorts, of which the first three were designated for histological analysis and the latter three were designated for snap-freezing of the brains for other analyses. The first five cohorts were used for behavioral evaluation. Four mice were omitted from all behavioral analyses due to severe repetitive turning behavior after HI (*n* = 4, see Table 2 for group allocation).

### 2.2. Behavioral Testing

Throughout the experiment, mice were handled and behavior was scored by experienced experimenters who were blinded for HI-status and diet. Behavioral tests assessing motor performance were performed during the light phase, while tests assessing cognitive parameters were performed during the dark phase.

#### 2.2.1. Accelerating Rotarod

At 8 weeks after HI surgery (P65-68), mice performed a Rotarod task to test motor coordination. The protocol was adapted from [35]. During the training phase, mice were habituated on the Rotarod (TSE Systems, Berlin, Germany) for 150 s at 5 rotations per minute (rpm) once a day on two consecutive days. If mice were unable to complete training of at least 100 s on training day 2, they were omitted from the analysis (*n* = 2/1 sham/HI). On two consecutive days after training, mice performed three trials per day, during which the Rotarod accelerated from 5 to 40 rpm over the course of 300 s. For each session, time was recorded until the mouse fell off the Rotarod. Mice were returned to their home cage for 30–45 min between trials. The mean latency to fall over all 6 trials was multiplied by a correction factor for weight, as a significant weight difference between diet groups was observed (see Section 3.1) and Mao et al. [36] previously showed that Rotarod performance in mice is strongly affected by bodyweight. Rotarod performance was thus calculated as follows: mean trial duration in seconds × individual mouse weight/mean weight of all mice (males and females separately).

#### 2.2.2. Novel Object Recognition Task

The novel object recognition task (NORT) was conducted at 10 weeks after HI surgery (P83). The NORT is used to test long-term memory and reflects hippocampal and perirhinal cortex function [37]. The NORT was conducted during the active phase, under red light conditions. Mice were habituated in a rectangular plexiglass cage (length × width × height = 560 × 330 × 200 mm) for 10 min on four consecutive days, starting at P79. On the fifth day, mice were placed in the same experimental chamber, which now contained two identical objects. Mice were left to explore both objects until 38 s of exploration time had been reached for a maximum of 10 min [38]. Mice were then returned to the home cage for one hour. During the testing phase, one object was replaced by a novel object and mice were left to explore both objects for 10 min. The two objects used as familiar and novel object were similar in material and size, but differed in color, texture and orientation. Specifically, one object was a cylinder made of four stacked blue 50 mL Falcon tube-caps (Corning Inc., Corning, NY, USA) and the other object was a yellow, 8-hole Duplo brick (The Lego Group, Billund, Denmark). The location of the novel object was randomized between trials. Mice that failed to explore the objects for at least 20 s during the novel object phase were omitted from the analysis (*n* = 4/4 sham/HI, [39]). Time spent exploring the objects during the NORT, i.e., orienting the nose toward the object with a 1–2 cm distance, was scored by an experienced observer using The Observer software (Noldus, Wageningen, The Netherlands). Novel object preference was calculated as (time spent with novel object/time spent with both objects) × 100 [39]. Mice were randomized to receive either the cylinder or the Duplo brick as familiar object. Because novel object recognition failed in the sham-operated control groups when the Duplo was provided as a familiar object, only data that were recorded using the cylinder as the familiar object are shown here.

#### 2.2.3. Modified Hole Board

At 12 weeks after HI injury (P93), mice were tested using the cognitive version of the modified hole board (mHB), adapted from [40]. Briefly, mice were placed in a grey PVC arena (50 × 50 × 50 cm) containing a 10-hole board made of dark grey PVC. All holes were scented with vanilla (dissolved in water 0.02%, Biomin Benelux, Uden, The Netherlands) and contained a piece of almond fixed underneath a grid. In three cylinders marked by a white PVC ring, a piece of almond was placed on the grid as a food reward. In the week prior to testing, mice received a piece of almond in their home cage on two consecutive days. Mice were tested during the dark phase under red light conditions. Mice were allowed to explore the arena until each of the three food rewards was collected, for a maximum of 5 min, four times per day on five consecutive days, and on two days the following week. For each session, time was recorded until mice collected all rewards. The mHB was cleaned with paper towels, water and odorless soap after each trial.

#### 2.2.4. Cylinder Rearing Task

At 4 and 15 weeks after HI (P37 and P113), mice performed the Cylinder rearing task [33,34,41] to determine forepaw preference. Strong preference for the right (unimpaired) forepaw indicates functional damage caused by the HI lesion. Cylinder rearing tasks were both conducted during the light phase. Mice were placed in a plexiglass cylinder (ø 7.5 by 15 cm) in front of two mirrors placed at a 90-degree angle. Mice were allowed to explore the cylinder for a maximum of 5 min until at least 10 full forepaw rearings had occurred. Mice that failed to perform 10 rearings within 5 min were omitted from the analysis (*n* = 1/1 sham/HI). Trials were recorded with a video-camera and later scored by an experienced, blinded observer. When a rearing occurred, the first weight-bearing paw to touch the cylinder-wall was scored (right, left or both). Additionally, “both” was scored when the other forepaw followed immediately after the first. The right (unimpaired) forepaw preference was calculated as (right–left)/(left + right + both) × 100%.

### 2.3. Histology and Immunohistochemistry

Histological outcome at 15 weeks after HI (P114) was assessed as the primary outcome measure. Adult mice were anesthetized by an intraperitoneal injection of 0.1 mL 20% pentobarbital. Mice were intracardially perfused using phosphate-buffered saline (PBS) followed by 4% paraformaldehyde (PFA). Perfused brains were post-fixed in 4% PFA for 24 h, dehydrated, embedded in paraffin and cut in 8 µm coronal sections at the level of CA1 of the hippocampus (bregma −1.8 mm). After rehydration, antigen retrieval and blocking, sections were incubated with Mouse-anti-microtubule associated protein 2 antibody (MAP2; 1:1000, Sigma-Aldrich, St. Louis, MO, USA) to stain for grey matter, or Rat-anti-myelin basic protein antibody (MBP; 1:500, Merck, Darmstadt, Germany) for white matter, followed by Horse-anti-Mouse biotin (1:100, Vector Laboratories, Burlingame, CA, USA) or Rabbit-anti-Rat biotin (1:400, Vector Laboratories), respectively. Binding was visualized with the Vectastain ABC kit (Vector Laboratories) and 3,3′-diaminobenzidine (DAB). Separate adjacent sections were stained with hematoxylin and eosin (H&E). Area measurements of the ipsilateral and contralateral hemispheres were performed by a blinded observer, and ipsilateral tissue loss was calculated as 1 − (ipsi/contra) × 100% for MAP2, MBP and H&E analyses.

For microglia and astrocyte analysis, coronal sections were incubated with Rabbit-anti-ionized calcium-binding adaptor protein-1 antibody (IBA1; 1:500, Wako Chemicals, Richmond, VA, USA) and Mouse-anti-glial fibrillary acidic protein antibody (GFAP; 1:100, OriGene, Rockville, MD, USA), followed by Goat-anti-Rabbit Alexa594 and Goat-anti-Mouse Alexa488, respectively (1:500, Invitrogen, Waltham, MA, USA) and 4′,6-diamidino-2-fenylindool (DAPI). Fluorescence images were obtained in the primary somatosensory cortex and directly adjacent to the lesioned area for each hemisphere at 40× (for IBA1) and 20× (for GFAP) magnification using an Axio Observer Z1 Microscope with Zen software (Carl Zeiss, Oberkochen, Germany). The GFAP-positive area was obtained by manually thresholding the GFAP signal using ImageJ software (US National Institutes of Health, Bethesda, MD, USA). Microglia count and area/cell was assessed in a semi-automated manner. Briefly, the IBA1-positive area was manually thresholded, and microglia were automatically selected using “Analyze particles” in ImageJ [42]. IBA1-positive cells were manually checked for DAPI-labelled nuclei, and false positives were omitted. The IBA1-positive area/cell was obtained by dividing the positive area of selected cells (including ramifications) by the number of selected cells, where a larger value corresponds to a more ramified cell, thus indicating non-activated microglia. All analyses were carried out by a blinded observer.

Synaptic density was assessed by staining with anti-MAP2 (1:500, Sigma-Aldrich) and anti-synaptophysin (1:400, Abcam, Cambridge, UK). Three pictures were taken from the cortex of each hemisphere at 40× magnification. Synapse area and intensity were assessed using manual thresholding in ImageJ.

### 2.4. Western Blot

At 15 weeks after HI (P114), mice were decapitated after anesthetization with 20% intraperitoneal pentobarbital, and brains were immediately extracted, divided into the ipsilateral and contralateral hemispheres of the cerebrum and the whole cerebellum, and snap-frozen in liquid nitrogen. Frozen brains were stored at −80 °C. Proteins were extracted from 25 mg of cerebral hemisphere tissue, and 20 µg of protein was separated on Mini-PROTEAN TGX gels (4–20%, Bio-Rad, Hercules, CA, USA) and transferred onto an Amersham Hybond polyvinylidene difluoride membrane (GE Healthcare Life Sciences, Chicago, IL, USA). The membranes were blocked and incubated overnight at 4 °C with primary antibodies (anti-synaptophysin, 1:4000, 5461, Cell Signaling Technology, Danvers, MA, USA, anti-syntaxin 3, 1:1000, ab4113, Abcam, post-synaptic density protein 95, 1:1000, MAB1596, Merck, β-actin, 1:15,000, 60008-1, Proteintech). The next day, membranes were washed and incubated with horseradish-peroxidase-conjugated secondary antibodies (1:2000, P0447, P0448, Dako, Glostrup, Denmark). Proteins were visualized using Amersham enhanced chemi-luminescence prime western blot detection reagent (GE Healthcare Life Sciences) and a ChemiDoc XRS+ (Bio-Rad). Optical density was analyzed using ImageJ, and data was normalized for β-actin content per lane. Protein expression was calculated relative to the contralateral hemisphere of sham-operated mice that received the control diet.

### 2.5. Thin Layer Chromatography

Cerebral and cerebellar samples were taken from frozen brain tissue and extracted using a modified version of the Folch Method [43]. Briefly, tissue was sonicated in a ratio of 2:1 chloroform/methanol and 0.05% butylated hydroxytoluene (BHT), shaken using a Vibramax and centrifuged. The supernatant was aliquoted into an amber glass vial, 0.9% sodium chloride was added and samples were centrifuged for phase separation. The upper phase was removed and the lower phase kept with the remaining solvents, slowly evaporating under nitrogen gas. The concentrated lipid extract was reconstituted in 4:1 chloroform/methanol and 0.05% BHT and stored at −20 °C.

Thin layer chromatography (TLC) was performed on the lipids using a protocol adapted from [44]. Briefly, TLC silica gel plates (60G, Merck) were pre-washed with 1:1 chloroform/methanol, dried and pre-treated with 0.9% boric acid in ethanol. A phospholipid standard mixture containing L-α-Lysophoshatidylcholine (Sigma-Aldrich), sphingomyelin (SM; Sigma-Aldrich), L-α-phosphatidylcholine (PC; Sigma-Aldrich), L-α-phosphatidylinositol (PI; Avanti Polar Lipids, Birmingham, AL, USA), 1,2-diacyl-sn-glycero-3-phospho-L-serine (PS; Sigma-Aldrich), and phosphatidylethanolamine (PE; Sigma-Aldrich) was prepared. Each standard was prepared at a final concentration of 4 mg/mL in 4:1 chloroform/methanol with 0.05% BHT. Phospholipid samples and standards were then deposited onto the demarcated concentration zone on the TLC plates and exposed to the mobile phase by placing the plate in a glass tank containing 30% chloroform, 35% ethanol, 7% water, 35% trimethylamine. After this phase was completed, plates were dried and sprayed with a 0.01% primuline in 6:4 acetone/water solution. Images were taken using a ChemiDoc XRS+ (Bio-Rad). Phospholipids were quantified using ImageJ. Phospholipid bands were identified as regions of interest (ROI), and the sum of all the pixel intensities in each ROI was measured and labelled as the raw integrated density.

### 2.6. Fatty Acid Analysis

Fatty acids were extracted from approximately 20 mg of frozen brain tissue from each hemisphere. Tissue was added to 50 volumes of ice-cold ultra-pure water, followed by 30 s of sonication, before storage at −80 °C. A known amount of C19:0 PC was added as an internal standard to 500 µL sample and subsequently extracted according to Bligh & Dyer [45]. The dichloromethane layer was evaporated to dryness and the extracted lipids were converted to fatty acid methyl esters (FAMEs) with methanol and 2% sulfuric acid at 100 °C for 60 min. The FAMEs were extracted with hexane and, after evaporation, dissolved in isooctane. 1 µL of the isooctane was injected into the gas chromatography (GC). FAMEs were separated on a CP-Sil 88 column and detected with a flame ionization detector (FID). FAME identification was based on retention time. The relative concentration was based on the peak area, and the absolute concentration was calculated after normalization with the C19:0 peak area. The absolute concentration is shown in mg/L and is used to calculate the total *n*-3, *n*-6 and *n*-9 fatty acids and the *n*-6/*n*-3 ratio.

### 2.7. Phospholipid-Bound Fatty Acids

A known amount of C19:0 PC was added as an internal standard to 400 µL sample (1:50 brain homogenate) and subsequently extracted according to Bligh & Dyer [45]. The phospholipid fraction was separated from the other lipid classes by Solid Phase Extraction (SPE) and subsequently converted to FAMEs with methanol +2% sulfuric acid at 100 °C for 60 min. The FAMEs were extracted with hexane and, after evaporation, dissolved in isooctane. 1 µL of the isooctane was injected into the GC. FAMEs were separated on a CP-Sil 88 column and detected with a FID detector. FAME identification was based on retention time. The relative concentration was based on the peak area, and the absolute concentration was calculated after normalization with the C19:0 peak. The absolute concentration is shown in mg/L.

### 2.8. Statistical Analysis

All statistical analyses were performed using R (version 3.5.2, R Foundation for Statistical Computing, Vienna, Austria). Data was analyzed using a full factorial three-way General Linear Model (GLM) with HI status (HI or sham), diet (experimental or control diet) and sex (male or female) as predictors, as well as a random factor for litter nested within the experimental cohort, using the ‘lme4′ package in R [46]. If no significant interactions of sex were found, the analysis was performed as a two-way GLM with HI status and diet as predictor variables, with random effects for litter number and cohort. After significant effects were found in the model, planned comparisons were carried out using the Holm correction, comparing HI and sham-groups of either diet, as well as HI groups within diets (family of three comparisons). If there were factors interacting with sex (two-way or three-way interactions), pairwise comparisons were performed separately for male and female mice. Repeatedly measured weight data was analyzed using a mixed model with the same predictors (HI, diet and sex) in addition to random effects for mouse ID, litter number and experimental cohort. In all analyses except for food intake, the experimental units were individual mice. Food intake was measured per litter (before weaning) or cage (after weaning), averaged over the number of experimental animals per cage, and the cage was used as the experimental unit in the analysis. Graphs were created using GraphPad Prism 8.3 (GraphPad Software, San Diego, CA, USA). Raw data is shown as mean ± standard error of the mean (SEM), sample sizes for each analyses are mentioned in the figure captions. Results were considered statistically significant if *p <* 0.05, trends were reported when *p <* 0.10.

## 3. Results

### 3.1. Food Intake and Bodyweight

No differences in food intake were found between groups on the experimental diet and control diet either before weaning (main effect of diet, n.s., Figure 1A) or after weaning (main effect of diet, n.s., Figure 1B,C). Mice that received the experimental diet had a higher bodyweight than mice on the control diet (main effect of diet, *p <* 0.01). Food intake was higher in males than in females (main effect of sex, *p <* 0.01) and males were heavier than females (main effect of sex, *p <* 0.001, Figure 1D,E). HI-injured mice were lighter than sham-operated mice in both diet groups (main effect of HI, *p <* 0.01).

### 3.2. Behavioral Parameters

At 4–15 weeks after induction of HIE, mice were subjected to a behavioral test battery to assess their motor and cognitive performance (Figure 2A). As there were no differences as a function of sex on the behavioral performances, data is shown for both sexes combined (Figure 2B–D). Behavioral data per sex are shown in Appendix A.

#### 3.2.1. Motor Performance

At 8 weeks after HI injury, HI-injured mice performed significantly worse on the accelerating Rotarod than sham-operated mice (main effect of HI, *p <* 0.001). Pairwise comparisons showed HI-mice performed significantly worse than sham-operated mice in the control diet group (*p <* 0.001) and in the experimental diet group (*p <* 0.05, Figure 2B). There were no differences between HI animals that were fed the control diet vs. those that were fed the experimental diet, indicating the experimental diet did not improve HI-induced motor deficits on the Rotarod.

At both 4 and 15 weeks after HI injury, HI-injured mice showed a significant preference for the unimpaired forepaw during the cylinder rearing task, indicating unilateral motor deficits (at 4 weeks after injury: main effect of HI, *p <* 0.01, data not shown, at 15 weeks after injury: main effect of HI, *p <* 0.001, Figure 2D). Pairwise comparisons confirmed that HI-injured mice had an increased preference for the unimpaired forepaw compared to their respective sham-control group (4 weeks after injury: experimental diet HI vs. sham, *p <* 0.001, control diet HI vs. sham, *p <* 0.01; 15 weeks after injury: experimental diet HI vs. sham, *p <* 0.001, control diet HI vs. sham, *p* < 0.001, Figure 2D). There were no differences between HI animals that were fed the control diet vs. those that were fed the experimental diet, indicating that the experimental diet did not improve HI-induced unilateral sensorimotor impairments in the cylinder rearing task.

#### 3.2.2. Cognitive Performance

A significant main effect of HI injury on the percentage of time spent exploring the novel object was observed at 10 weeks after induction of HI (*p <* 0.01). Pairwise comparisons showed that HI-injured mice that were fed the control diet spent significantly less time with the novel object than sham-operated mice (*p <* 0.01), whereas this difference was not significant for HI-injured mice on the experimental diet (*p <* 0.10, Figure 2C). Furthermore, HI-injured mice that were fed the experimental diet tended to perform better than those on the control diet, although this difference was not significant (*p <* 0.10). Importantly, all experimental groups except the control diet HI-injured group had a significant preference for the novel object (control diet + sham: 70.9% vs. 50%, *p <* 0.001, control diet + HI: 54.1% vs. 50%, *p* > 0.10, experimental diet + sham 69.6% vs. 50%, *p <* 0.001, experimental diet + HI: 63.6% vs. 50%, *p <* 0.001), indicating that recognition memory after HI was improved by the experimental diet.

In addition to the NORT, mice also performed the mHB task to assess cognitive performance. This was based on the motivation to obtain a food reward; however, in hindsight, performance on this test could be influenced by the diet (e.g., [47]). The results of this task were therefore deemed open to potential bias and thus inconclusive, and are shown in Appendix A.

### 3.3. Histology

#### 3.3.1. Ipsilateral Tissue Loss

A significant tissue loss was observed after HI injury, as reflected by the ipsilateral tissue loss quantified using H&E staining (main effect of HI, *p <* 0.001, Figure 3A,B). Furthermore, an interaction effect between HI injury and diet (*p <* 0.05) and a three-way interaction between HI injury, diet and sex (*p <* 0.05) were found. Because of the interactions with sex, pairwise comparisons were carried out for males and females separately. Post-hoc tests showed that ipsilateral tissue loss was significantly reduced in HI males that were fed the experimental diet compared to those that were fed the control diet (*p <* 0.01, Figure 3B). An effect of the experimental diet was not observed in HI-injured females.

#### 3.3.2. Ipsilateral Grey Matter Loss

Ipsilateral grey matter as quantified by MAP2-positive staining was significantly affected by HI injury, diet and sex (main effect of HI, *p <* 0.001, interaction HI × diet, *p <* 0.05, interaction HI × sex, *p <* 0.10, interaction HI × diet × sex, *p <* 0.05, Figure 3A,C). Pairwise comparisons were subsequently carried out for males and females separately and showed that ipsilateral grey matter loss was significantly reduced in HI-injured males that were fed the experimental diet compared to the control diet (*p <* 0.01, Figure 3C), whereas no significant effect of the experimental diet was found in females.

#### 3.3.3. Ipsilateral White Matter Loss

Ipsilateral white matter as quantified by MBP-positive staining was also significantly affected by HI injury, diet and sex (main effect of HI, *p <* 0.01, interaction HI x sex, *p <* 0.10, interaction HI × diet × sex, *p <* 0.05, Figure 3A,D). Pairwise comparisons carried out for both sexes separately showed that white matter loss in the ipsilateral hemisphere was reduced in HI-injured males that were fed the experimental diet compared to males fed the control diet (*p <* 0.05).

### 3.4. Neuroinflammation

#### 3.4.1. Microglial Activation

Microglial activation in the ipsilateral hemisphere was assessed by IBA1 immunofluorescent staining in the cortex and perilesional area (Figure 4A,B). The number of IBA1/DAPI-positive cells was increased in both areas in HI-injured mice (cortex: main effect of HI, *p <* 0.01, perilesional area: main effect of HI, *p <* 0.01). There were trending interactions between HI injury and sex (cortex: interaction HI × sex, *p* = 0.097, perilesional area: interaction HI × sex, *p* = 0.051), therefore pairwise comparisons were carried out for both sexes separately. Post-hoc tests showed that the number of IBA1/DAPI-positive cells was significantly higher in HI-injured males that had been fed the control diet compared to sham-operated controls, but not those that were fed the experimental diet (cortex: control diet HI vs. sham, *p <* 0.01, Figure 4B,C, perilesional area: control diet HI vs. sham, *p <* 0.05, Figure 4F). In females, no significant differences were observed in the IBA1-positive cell count between the HI-injured and sham-operated groups, nor between the diet groups.

The cell size of IBA1-positive cells was quantified for IBA1/DAPI-positive cells and was significantly reduced by HI injury (cortex: main effect of HI, *p <* 0.01, perilesional area: main effect of HI, *p <* 0.01), which is indicative of a pro-inflammatory, amoeboid phenotype. Because of the interaction involving sex on the number of IBA1-positive cells, pairwise comparisons were done for both sexes separately. These comparisons revealed that the IBA1-positive cell size was significantly decreased after HI injury in male mice that were fed the control diet (cortex: control diet HI vs. sham, *p <* 0.01, Figure 4D, perilesional area: control diet HI vs. sham, *p <* 0.05, Figure 4G), whereas there were no differences between HI and sham in males that were fed the experimental diet, or in any of the female groups (Figure 4C,F). This indicates that the experimental diet reduced HI-induced microglia activation.

#### 3.4.2. Astrocyte Reactivity

HI had a significant effect on astrocyte reactivity, as quantified by the GFAP-positive area in the ipsilateral cortex (main effect of HI, *p <* 0.001, Figure 4B,E). There were several interactions involving HI, diet and sex (interaction HI x diet, *p <* 0.01, interaction HI × sex, *p <* 0.05, interaction effect HI × diet × sex, *p <* 0.05). To determine the origin of these effects, pairwise comparisons were performed for both sexes separately. For males on the control diet, the GFAP-positive area was increased after HI compared to sham (cortex: *p <* 0.001, perilesional area: *p <* 0.01). Importantly, GFAP reactivity was reduced in the cortex of HI-injured males that were fed the experimental diet compared to the control diet (*p <* 0.001). In females, no significant effects of HI on cortical GFAP were found for either diet group (control diet HI vs. sham, *p <* 0.10, experimental diet HI vs. sham, *p <* 0.10). In the area surrounding the lesion, GFAP reactivity was seen in all HI-groups compared to controls (main effect of HI, *p <* 0.001), and no significant effects involving diet or sex were found (Figure 4H).

### 3.5. Synaptic Markers

Western Blot was used to assess the cerebral protein levels of the synaptic markers syntaxi*n*-3, synaptophysin and post-synaptic density protein 95 at 15 weeks after HI. No differences between HI and sham mice and/or diet groups were found in the ipsilateral hemisphere for these markers (data not shown). Similarly, synaptophysin staining was assessed by immunofluorescence in the ipsilateral cortex, but no differences between experimental groups were found (data not shown).

### 3.6. Fatty Acids

No effects involving sex were found for fatty acid levels, therefore data is shown for males and females combined (Figure 5A–G). Fatty acid data divided by sex is shown in Appendix A. All measured fatty acid species are summarized in Appendix A. Fatty acid levels measured in the contralateral hemisphere are shown in Appendix A.

#### 3.6.1. Fatty Acid Ratios

The ratio of *n*-6 to *n*-3 PUFAs in the ipsilateral hemisphere of the brain was significantly decreased by the experimental diet in both HI and sham groups, indicating a higher presence of *n*-3 to *n*-6 PUFAs (main effect of diet, *p <* 0.001, Figure 5A). The ratio of *n*-6/*n*-3 was significantly lower in both HI-injured (*p <* 0.001) and sham-operated mice (*p <* 0.001) that were fed the experimental diet compared to mice that were fed the control diet (Figure 5A). There was no effect of HI on the *n*-6/*n*-3 ratio.

The total *n*-3 PUFA content was decreased in HI-injured mice (main effect of HI, *p <* 0.05), while it was slightly increased by the experimental diet (main effect of diet, *p <* 0.10, Appendix A). Total *n*-6 PUFAs were less abundant in mice that were fed the experimental diet (main effect of diet, *p <* 0.001, Appendix A), and in HI-injured mice that were fed the control diet (interaction effect HI × diet, *p <* 0.05).

No significant differences between groups were found for the total amount of the *n*-3 fatty acid α-linolenic acid (ALA) or the *n*-6 fatty acid linoleic acid (LA, Appendix A).

#### 3.6.2. EPA (20:5*n*-3)

The amount of EPA was significantly increased in the ipsilateral hemisphere of HI-injured mice (main effect of HI, *p <* 0.001) and in mice that were fed the experimental diet (main effect of diet, *p <* 0.001). There was a significant interaction between HI injury and diet (*p <* 0.01). Pairwise comparisons showed both a significant increase in cerebral EPA in HI-injured (*p* < 0.001) and sham-operated (*p* < 0.001) mice fed the experimental diet compared to those that were fed the control diet, and between the HI-injured and sham animals within the experimental diet group (*p <* 0.001, Figure 5B). There was no difference between the HI-injured and sham-operated mice in the control diet group.

More EPA was incorporated into phospholipids in the ipsilateral cerebrum of mice that had been fed the experimental diet (main effect of diet, *p <* 0.001). Pairwise comparisons confirmed a significant difference between HI groups (*p <* 0.001) and sham groups (*p* < 0.001) that were fed the experimental diet compared to the control diet (Figure 5C). Although no main effect of the HI injury was found, pairwise comparisons revealed a significant difference between HI and sham in the experimental diet group (*p <* 0.01), but not in the control diet group.

#### 3.6.3. DPA (22:5*n*-3)

The amount of docosapentaenoic acid (DPA, 22:5*n*-3) was significantly increased in the ipsilateral hemisphere of mice that received the experimental diet (main effect of diet, *p <* 0.001) and mice that had HI injury (main effect of HI, *p <* 0.05). Pairwise comparisons showed that the experimental diet increased *n*-3DPA in both the sham-operated group (*p <* 0.001, Figure 5D) and HI-injured group (*p* < 0.001) compared to their relative control diet groups. Furthermore, *n*-3DPA was significantly increased in response to HI injury in the experimental diet group (*p <* 0.05).

The *n*-3DPA incorporated into phospholipids was significantly affected by diet (main effect of diet, *p <* 0.001). Pairwise comparisons confirmed that HI-injured mice fed the experimental diet had more *n*-3DPA incorporated into phospholipids than HI-injured mice that had been fed the control diet (*p <* 0.001, Figure 5E). The same was found for sham-operated groups (*p <* 0.001).

#### 3.6.4. DHA (22:6*n*-3)

The amount of DHA in the ipsilateral hemisphere was decreased in HI-injured mice (main effect of HI, *p <* 0.05), which was shown in both the control diet (sham vs. HI, *p <* 0.01, Figure 5F) and to a lesser extent in the experimental diet group (sham vs. HI, *p <* 0.10). There was a trending effect suggesting that the experimental diet increased DHA levels (main effect of diet, *p <* 0.10). Pairwise comparisons showed that the experimental diet increased DHA content in both sham-operated (*p <* 0.10) and HI-injured mice (*p <* 0.01).

The DHA incorporated into phospholipids was increased in mice that were fed the experimental diet compared to the control diet (main effect of diet, *p <* 0.05, Figure 5G). The proportion of bound DHA was also influenced by the HI injury (main effect of HI, *p <* 0.01). Pairwise comparisons showed that significantly less DHA was phospholipid-bound in HI compared to sham animals that were fed the control diet (*p <* 0.05), whereas this HI-induced decrease of bound DHA was not observed in mice that were fed the experimental diet (*p <* 0.10). Importantly, HI-injured mice that were fed the experimental diet had a larger proportion of DHA incorporated into phospholipids than HI-injured mice that were fed the control diet (*p <* 0.01, Figure 5G). This effect was also found for sham-operated mice (*p <* 0.05).

### 3.7. Phospholipids

There were no significant differences between HI and sham, diet groups and sexes in the level of phosphatidylcholine (PC), phosphatidylethanolamine (PE), phosphatidylinositol (PI), phosphatidylserine (PS), sphingomyelin (SM) and the total amount of phospholipids per mg protein in the ipsilateral hemisphere of the cerebrum (Appendix A), the contralateral hemisphere, or in both hemispheres of the cerebellum (data not shown).

## 4. Discussion

HI is a prevalent cause of neonatal morbidity, leading to lifelong motor and cognitive impairments. There is a pressing need for new therapies that can protect or repair the injured neonatal brain, as hypothermia within 6 h after birth is currently the only option clinically available. Nutritional interventions could be implemented relatively rapidly and safely into the clinic and are therefore a potential next step to take towards reducing the devastating consequences of HIE. The aim of this study was to test the efficacy of an experimental concept diet, containing DHA, EPA, UMP, choline, iodide, zinc, and vitamin B12, in a mouse model of perinatal HIE. The listed nutrients provide building blocks for neuronal and synaptic membranes, potentially acting synergistically to support brain development by improving synaptic connectivity, reducing neuroinflammation, and enhancing neuronal survival after injury [10,11,48,49,50,51].

The current study demonstrates that the experimental diet reduced HI lesion size in males. It has long been known that males are more susceptible than females to develop lasting consequences from HIE [1]. As reviewed by Hill & Fitch [52], the sexual dimorphism seen in humans is also found in rodent models of HIE, such as the Vannucci–Rice model used in this study. Indeed, ipsilateral tissue loss in the control diet group was ~51% in males and ~35% in females at 15 weeks post-HI, and the experimental diet was effective in reducing lesion size (to ~32%) in males only. Previously, several factors have been suggested to explain higher susceptibility to injury in males, among which pre-pubescent hormonal differences, genetic predisposition, larger size at birth, preferential activation of different apoptotic pathways [53], and sex-specific immune activation [4,52]. In this study, evidence was found to suggest that the experimental diet reduced lesion size in males by dampening neuroinflammation, which is exacerbated in males after HI injury, although parallel working mechanisms cannot be ruled out.

### 4.1. Neuroinflammation

Inflammation in the brain after HI is a key mediator of brain injury. Microglia respond after the acute phase by switching to a pro-inflammatory state, increasing in number and attracting peripheral immune cells to the site of injury by pro-inflammatory cytokine and free radical release [5,8]. Once there, both microglia and peripheral immune cells release additional factors that are neurotoxic or prevent regeneration of neurons and axons. In this study, we have shown that the number of IBA1-positive cells, a marker for microglia/macrophages, was increased in the area proximal to the lesion and in the distal cortical areas in HI-injured males. In addition, IBA-positive cells were smaller, indicating less ramifications and a more active, amoeboid-shaped and pro-inflammatory microglial phenotype [3]. Females are in general less affected by immune activation in the acute and chronic stages of HIE, which may be caused by reduced cytokine levels, microglial activation and infiltration of peripheral immune cells compared to males [4,5,54]. Therefore, we hypothesize that the experimental diet was especially effective in reducing the more prominent neuroinflammatory response observed in males.

The anti-inflammatory properties of *n*-3 fatty acids are well-established, and they encompass (among others) the downregulation of the pro-inflammatory NF-κB pathway, thereby inhibiting the expression of pro-inflammatory cytokines, whilst functioning as precursors to anti-inflammatory lipid mediators and reducing damage caused by reactive oxygen species (ROS, [51,55]). The increased *n*-3 fatty acids and decreased *n*-6/*n*-3 fatty acid ratio found in the brain of mice fed the experimental diet may have thus dampened neuroinflammation after HI injury, diminishing secondary damage and reducing lesion size. The observations on neuroinflammation in this study are in agreement with a study in mice subjected to transient middle cerebral artery occlusion, a model for adult stroke, in which supplementation of a similar synergistic diet concept containing DHA, EPA, UMP and choline (among other components), reduced the number of IBA1-positive cells and microglia activation measured by PET in male mice [50]. Whether the experimental diet was equally effective in reducing acute and chronic neuroinflammation after neonatal HIE remains subject to further investigation.

Astrocytes, another crucial glial cell type, respond to HI injury in a process called “reactive astrogliosis”, which leads to increase of astrocyte numbers, exacerbation of neuroinflammation and formation of glial scarring [56], a process which can take place weeks to months after the initial injury has occurred [8]. In this study, astrogliosis was enhanced in HI-injured males in the cortex, whereas this effect was substantially reduced by the experimental diet. The glial reaction is proposed to be beneficial when it acutely surrounds the lesioned area, thereby preventing further tissue damage, but detrimental in the long term as a mature persistent scar can prevent axonal regeneration after injury [56].

Together, these findings indicate that the experimental diet may modulate the neuroinflammatory response, that is more severe and persistent in males after HIE, thereby reducing secondary damage and/or facilitating regeneration.

### 4.2. Phospholipids & Fatty Acids

Through a cascade of biochemical reactions known as the Kennedy cycle, DHA, UMP and choline are used to synthesize the phospholipids PC and PE [10,11]. PC is the most abundant in the brain as a component of the phospholipid bilayer. It was hypothesized that dietary supplementation of these precursors would enhance synthesis of neural and synaptic components [11], thereby possibly contributing to neurogenesis or neurorepair in HI-injured mice. In the current study, continuous exposure to the experimental diet starting from birth onwards did not result in altered brain phospholipids produced by the Kennedy cycle (PC or PE) and ARA-containing phospholipids (i.e., PI) at 15 weeks of after HI, indicating that alteration of phospholipid species may not be underlying the beneficial effects of the experimental diet found in this study. Previous studies showed alterations in brain phospholipid profiles after 42 days of choline supplementation in adult male rats [57], supplementation for 70 days with a similar multi-nutrient in adult mice undergoing traumatic brain injury [58], and in rats after supplementation with *n*-3 fatty acids from the second day of gestation until P14 [22]. Further studies are required to assess the potential influence of different life stages and exposure duration on the ability of dietary precursors to affect phospholipid synthesis and incorporation in neuronal membranes.

Although phospholipid species were not significantly changed, it was found that the *n*-3 fatty acids EPA, *n*-3DPA and DHA were increased in the brain phospholipids of mice that were fed the experimental diet. Increased *n*-3 PUFA levels in neuronal membranes have been reported to facilitate fluidity of neuronal membranes, thereby leading to enhanced signaling [51,59]. Furthermore, a proportion of phospholipid-bound *n*-3 fatty acids can be released and converted to oxylipins, such as neuroprotectins and resolvins, that help resolve neuroinflammation and prevent apoptosis after injury [55,59,60]. Interestingly, DHA was decreased in the phospholipids of HI-injured brains compared to controls, whereas phospholipid-bound EPA, total EPA and total *n*-3DPA was increased after HIE. Both a decrease in brain DHA and an increase in *n*-3DPA have been reported previously in a rat model of HI injury [61]. Authors hypothesized that this was due to impaired peroxisomal β-oxidation, a process that is required to convert *n*-3DPA or EPA to DHA [61,62]. Impaired β-oxidation may explain the accumulation of *n*-3DPA and EPA and the reduction of DHA both in free form and bound to phospholipids in HI-injured brains. Hence, increasing dietary supply of preformed DHA may have enhanced neurorepair after HI injury in this study.

Although not directly addressed in this study, involvement of other pathways through which the experimental diet reduced lesion size after HI injury cannot be ruled out. For example, *n*-3 fatty acids, zinc and iodine have been known to reduce apoptosis, either directly through stimulating neuronal survival or through reduction of ROS [21,22,25,26]. Furthermore, this study was not designed to investigate effects of individual components, nor to identify potential synergy between combined dietary components. Therefore, the extent to which each of the individual dietary components contributed to the effects reported here remains subject to further investigation.

### 4.3. Functional Outcome

Perinatal HI injury leads to lifelong impairments across the motor and cognitive domains, including an increased risk for developing cerebral palsy [2]. The NORT has been shown to rely greatly on the hippocampus and perirhinal cortex [37,38] and performance is impaired in rodent models of HI injury [33,63,64]. Object preference towards exploring the novel object was not observed (50% novel, 50% familiar) in the HI-injured control diet group, indicating impaired cognition, whereas novel object preference was at sham-levels for the HI-injured group exposed to the experimental diet. A reduced lesion size in the hippocampus and perirhinal cortex may have improved functional outcomes in mice that were fed the experimental diet. In preliminary clinical studies, dietary supplementation with similar nutrients as used in the current study was found to generate clinically relevant improvements in Bailey III cognitive and language scales in infants with cerebral palsy [15], and those at risk of developing neurodevelopmental impairments [28]. Moreover, children with perinatal brain injury that received these dietary supplements showed significantly enhanced attention scores at 4–5 years of age [65]. Although these findings have to be replicated in larger cohorts, the combination of these studies and what was found here in an experimental model support the hypothesis that the nutrients in the experimental diet leads to reduced lesion size and better functional outcome in the cognitive domain. In addition to the NORT, animals in this study were also subjected to the mHB, which is recommended as a test for several cognitive domains such as short-term and long-term memory [40]. The data obtained in this food-motivated functional test was in hindsight deemed inconclusive, mainly because the motivation to obtain a food reward and overall satiety was potentially influenced by the dietary interventions (e.g., [47]). In addition, the animals were not fasted prior to conducting the task. For the sake of transparency, the mHB data are shown in Appendix A.

Although improved memory was found after the experimental diet in HI-injured mice, no such improvements were seen in motor function on two different tests taken at 4, 8 and 15 weeks after HI injury. In the clinical trials cited above, dietary supplementation also did not lead to an improved Bayley III motor performance [15,28], although these results need to be interpreted with caution due to the small sample size. From these results, it can be speculated that the experimental diet does not specifically target motor development or may be unable to markedly reduce the large lesion across the motor- and somatosensory areas caused by HI injury. Furthermore, motor development after HI injury is enhanced by environmental enrichment (e.g., [66]), which was not provided to the animals in this study.

Weight gain is an important indicator of growth during the neonatal period, as it signifies better overall growth and head circumference, in turn enabling brain development [9]. Although pups of both diet groups had the same weight around birth, those that were subjected to HI injury on P9 had a slower growth trajectory, which may indicate impaired development. Altered growth patterns have also been reported in children suffering from cerebral palsy [67,68]. In the current study, HI-injured mice that received the experimental diet gained more weight than HI-injured control diet mice, which may indicate a more favorable growth trajectory. In a clinical setting, *n*-3 supplementation has been shown to increase weight at discharge and head circumference z-scores in very low birth weight infants (<1500 g, [69]). Together, these results may indicate that the experimental diet used in the current study supports functional weight gain, although future studies should assess body length or lean body mass to give a more accurate indication thereof.

### 4.4. Limitations

An important limitation of this study is that, based on the literature and previous experiments performed by our group, sex-dependent effects after exposure to the diet were not expected and sex was not included as a factor in the sample size calculations. However, this study showed that the effects of the diet on lesion size and neuroinflammatory markers were different for males and females. Therefore, these results were analyzed using a 2 × 2 × 2 design (HI status × Diet × Sex) instead of an anticipated 2 × 2 design (HI status x Diet), which resulted in reduced statistical power. Repeating this experiment with more power to detect interactions involving sex could give more insight into the working mechanism of the diet.

The dietary intervention was provided from birth throughout life, whereas HIE was induced at P9. Therefore, it is impossible to definitively conclude whether the diet was protective or regenerative in nature, or both. In particular, possible transfer of specific nutrients via dam’s milk in the days preceding the insult could have reduced the extent of the injury by dampening the immune response and/or reducing the effects of oxidative stress, while the same nutrients could also have helped regenerate neuronal loss and aid brain development at later stages [55]. Discovering the primary working mechanism of the experimental diet could have several implications for clinical practice. If the diet is mainly protective, the start of the intervention should be early and even preventive treatment could be applied, for example in infants born preterm. If the diet mainly enhances regeneration, it may stretch the therapeutic window beyond the ~3–6 h after birth that is currently effective for hypothermia [8]. Further studies should expand on this data by studying the efficacy of the dietary intervention within several timeframes and durations.

Due to the early start of the intervention, it was decided to provide the experimental diet to the mother, thereby depending on the transfer of nutrients through milk in the first few weeks of life. In this study, the neonatal period was arguably the most important therapeutic window, as it both occurred around the injury and in the period of life where the brain is expanding rapidly. For some nutrients such as DHA, EPA, iodide, and vitamin B12, milk levels are known to depend on maternal diet and stores [70,71], but for others (UMP, zinc), it is unknown whether maternal supplementation increases its availability in milk. Furthermore, other effects of the experimental diet on milk composition (other than the supplemented ingredients) and/or maternal care or behavior cannot be ruled out. In clinical practice, the intervention can be given directly to the infants through (par)enteral feeding, potentially increasing the efficacy of the experimental diet.

## 5. Conclusions

The data from the current study indicate that exposure to an experimental diet containing additional DHA, EPA, UMP, choline, iodide, zinc, and vitamin B12 from birth onwards may reduce HI-induced lesion size and neuroinflammation in male but not in female mice. The sex-dependent severity of injury may underlie this difference in treatment efficacy. In both sexes, the experimental diet led to better functional outcome in the cognitive domain. These beneficial effects may be explained at least in part by increased levels of total and phospholipid-bound *n*-3 fatty acids in neural membranes, or specifically by the restoration of HI-induced depletion of cerebral DHA levels. Future studies are needed to further assess the efficacy of the (combined) nutrients in the experimental diet and the underlying neuroprotective/regenerative mechanisms after perinatal HIE.

## Figures and Tables

**Figure 1 nutrients-14-00176-f001:**
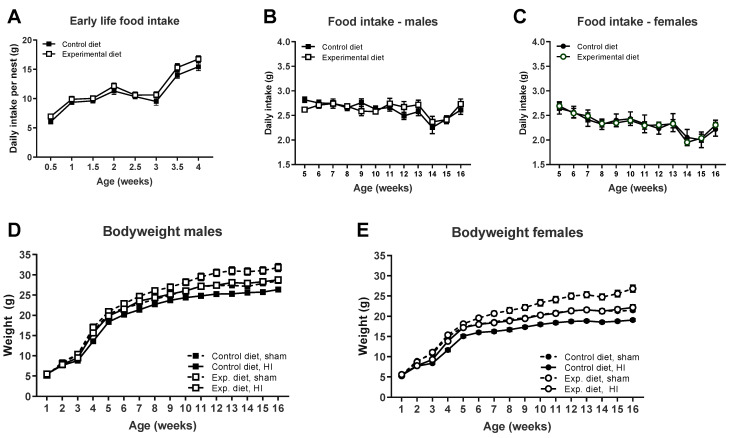
Food intake was similar between mice that were fed the experimental diet or the control diet either before weaning ((**A**), *n* = 12 litters per diet) or after weaning ((**B**), males: control diet *n* = 13 cages, experimental diet *n* = 11 cages; (**C**), females: control diet *n* = 12 cages, experimental diet *n* = 11 cages). Mice that were fed the experimental diet weighed more than mice that received the control diet (males: HI-injured control diet vs. experimental diet, *p <* 0.01; females: HI-injured control diet vs. experimental diet, *p <* 0.01), and HI-injured animals weighed less than sham-operated controls (males: control diet HI vs. sham, *p <* 0.01, experimental diet HI vs. sham, *p <* 0.01; females: control diet HI vs. sham, *p <* 0.001, experimental diet HI vs. sham, *p <* 0.001). (**D**,**E**): control diet, sham, *n* = 16/16 male/female, control diet, HI, *n* = 16/17, experimental diet, sham, *n* = 16/19, experimental diet, HI, *n* = 18/18.

**Figure 2 nutrients-14-00176-f002:**
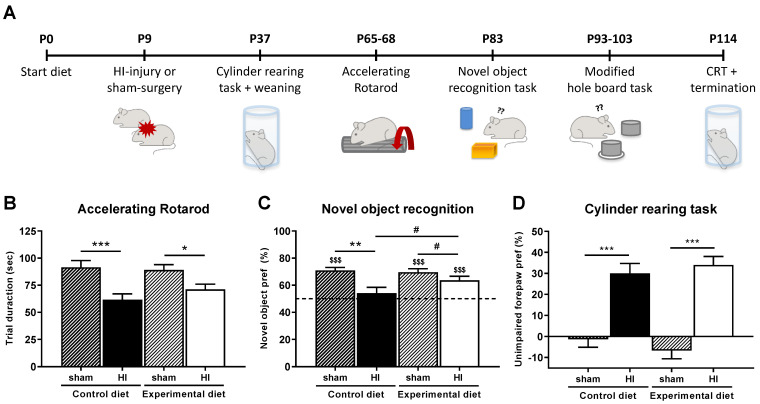
Schematic overview of the experimental timeline (**A**). The experimental diet did not improve HI-induced motor impairments on the accelerating Rotarod ((**B**), control diet: sham, *n* = 25, HI, *n* = 25, experimental diet: sham, *n* = 27, HI, *n* = 28). HI injury impaired novel object recognition in mice fed the control diet, but not in mice fed the experimental diet ((**C**), control diet: sham, *n* = 10, HI, *n* = 11, experimental diet: sham, *n* = 14, HI, *n* = 14). The experimental diet did not reduce HI injury induced impairments in unilateral sensorimotor impairment on the cylinder rearing task (P114 shown in (**D**), control diet: sham, *n* = 26, HI, *n* = 21, experimental diet: sham, *n* = 28, HI, *n* = 27). # corrected *p <* 0.10, * corrected *p <* 0.05, ** corrected *p <* 0.01, *** corrected *p <* 0.001, $$$ *p <* 0.001 compared to 50% novel object preference, P = postnatal day, HI injury = hypoxic-ischemic injury.

**Figure 3 nutrients-14-00176-f003:**
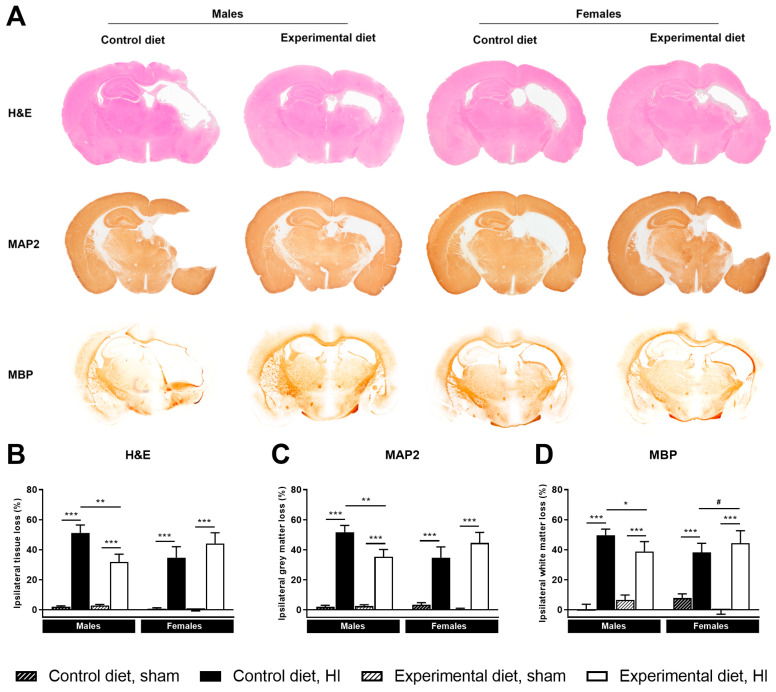
Representative images of coronal brain sections that were analyzed for lesion size (H&E), grey matter loss (MAP2) and white matter loss (MBP) for both diets after HI injury (**A**). Due to the effect of sex, males and females are depicted separately. Lesion size (**B**), grey matter loss (**C**) and white matter loss (**D**) were reduced in HI-males that were fed the experimental diet compared to males fed the control diet. (**B**–**D**): control diet, sham, males, *n* = 10/10 male/female, control diet, HI, *n* = 10/10, experimental diet, sham, *n* = 9/12, experimental diet, HI, *n* = 10/9, # corrected *p <* 0.10, * corrected *p <* 0.05, ** corrected *p <* 0.01, *** corrected *p <* 0.001.

**Figure 4 nutrients-14-00176-f004:**
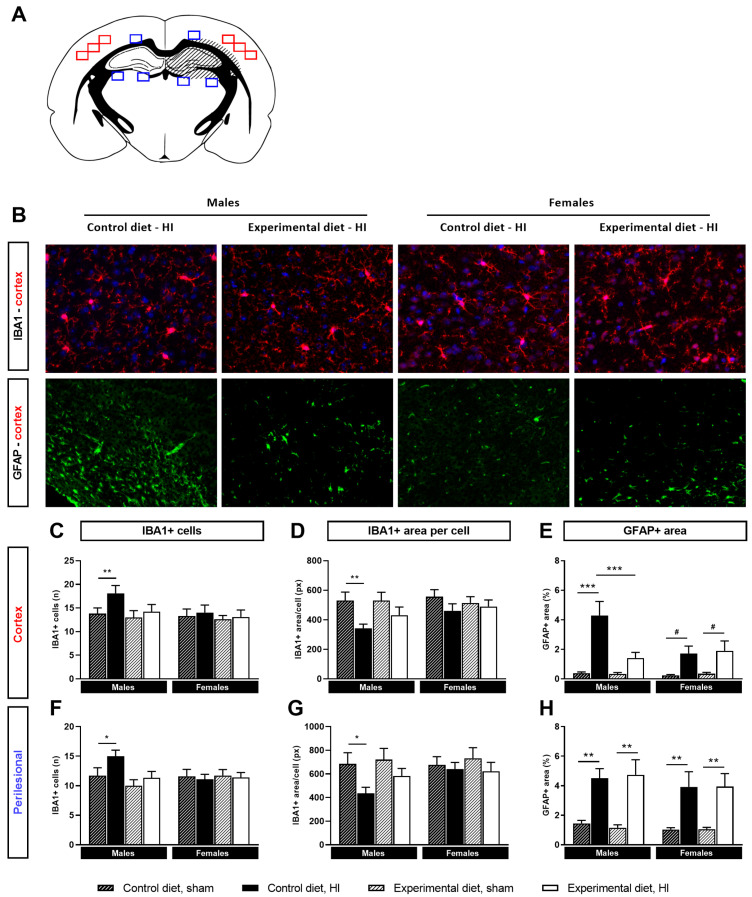
Markers for neuroinflammation were assessed in the cortex (red squares) and perilesional area (blue squares) for HI-injured and sham-operated mice fed the control or experimental diet (**A**). Representative images taken in the cortex were shown for IBA1 (40× objective) and GFAP (20× objective, (**B**)). Because of the effect of sex, data for males and females are depicted separately (**B**–**H**). IBA1/DAPI-positive cells were increased in the ipsilateral hemisphere of HI-injured males that were fed the control diet, both in the cortex (**C**) and perilesional area (**F**). Additionally, increased microglial activation, assessed by reduced IBA1-positive area per cell, was observed in HI-injured males that were fed the control diet compared to sham-operated males, whereas this was not found for HI-injured males that were fed the experimental diet (**D**,**G**). More astrocyte reactivity was seen in the cortex (**E**) and perilesional area (**H**) of HI-injured mice, and this was reduced in the cortex of males that were fed the experimental diet compared to the control diet (**D**). (**C**–**H**): control diet, sham, *n* = 10/10 male/female, control diet, HI, *n* = 10/9, experimental diet, sham, *n* = 8/11, experimental diet, HI, *n* = 9/10, # corrected *p <* 0.10, * corrected *p <* 0.05, ** corrected *p <* 0.01, *** corrected *p <* 0.001.

**Figure 5 nutrients-14-00176-f005:**
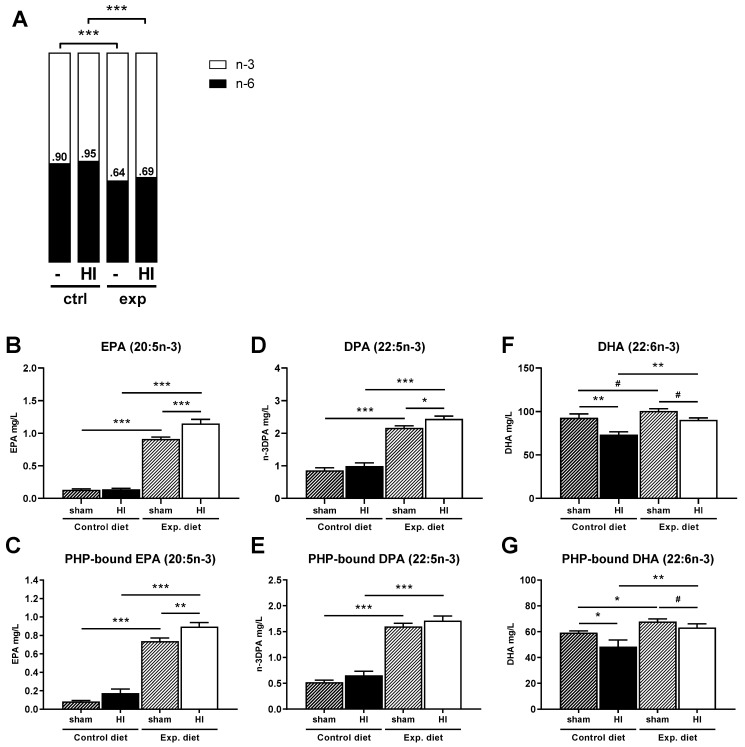
The ratio of *n*-6/*n*-3 fatty acids was decreased in the ipsilateral cerebral hemisphere of mice fed the experimental compared to those fed the control diet (**A**). The percentages of *n*-3 of total fatty acids EPA (**B**), *n*-3DPA (**D**) and DHA (**F**) were increased in mice fed the experimental diet. Levels of EPA (**B**) and *n*-3DPA (**D**) were also enhanced in HI-injured mice compared to sham-operated mice, while DHA was decreased in HI-injured mice (**F**). A higher ratio of *n*-3 fatty acids incorporated into phospholipids was found for EPA (**C**), *n*-3DPA (**E**) and DHA (**G**) in mice fed the experimental diet. (**B**,**D**,**F**): control diet: sham, *n* = 10, HI, *n* = 12, experimental diet: sham, *n* = 14, HI, *n* = 13, (**C**,**E**,**G**): control diet: sham, *n* = 12, HI, *n* = 6, experimental diet: sham, *n* = 13, HI, *n* = 8, # corrected *p <* 0.10, * corrected *p <* 0.05, ** corrected *p <* 0.01, *** corrected *p <* 0.001, ctrl = control diet, exp = experimental diet.

**Table 1 nutrients-14-00176-t001:** Ingredients (g/100 g), energetic value and fatty acids of the experimental diet and isocaloric control diet used in this study.

Ingredients	Control Diet	Experimental Diet
Carbohydrates		
Cornstarch, pre-gelatinized	27.47	25.11
Maltodextrin	15.50	15.50
Sucrose	10.00	10.00
Dextrose	10.00	10.00
Cellulose powder	5.00	5.00
Proteins		
Casein	20.00	20.00
Fats		
Soy oil	2.66	2.00
Coconut oil	1.26	0.10
Corn oil	3.08	1.70
Fish oil (HiDHA 25N + EPA)	-	3.20
Vitamins & Minerals		
Mineral & trace element premix (AIN-93G-MX)	3.50	3.50
Vitamin mix (AIN-93-VX)	1.00	1.00
Supplemented Vit. B12 (Cyanocobalamin 0.1%)	- ^1^	0.0125
Supplemented Zinc	- ^1^	0.1269
Supplemented Iodide	- ^1^	0.0080
Additions		
L-cystine	0.300	0.300
Choline chloride (0.434g/g)	0.230	0.922
Tert-butylhydroquinone	0.0014	0.0014
Soy lecithin (Emulpur)	-	0.7547
UMP disodium (24% H_2_O)	-	0.5000
Cytidine 5MP free acid	-	0.2634
Energetic value (kcal/100g)	384.1	374.6
% saturated fatty acids	26.55	26.10
% *n*-3 of total fatty acids	2.54	22.44
% *n*-6 of total fatty acids	46.82	26.72
*n*-6/*n*-3 ratio	18.53	1.19

^1^ This nutrient is present in the AIN93G vitamin mix or mineral and trace element premix but has not been additionally supplemented to the control diet. DHA = docosahexaenoic acid, EPA = eicosapentaenoic acid, *n*-6 = omega-6, *n*-3 = omega-3.

**Table 2 nutrients-14-00176-t002:** Number of animals used for this study for histology, molecular analyses and behavior, outliers and litter composition.

	Control Diet		Experimental Diet	
Group	Behavior	Histo|Mol	Behavior	Histo|Mol
HI-injured	29	20|13	31	20|16
Males	14 (1)	10|6	16 (1)	10|8
Females	15 (1)	10|7	15 (1)	10|8
Sham-operated	26	20|12	28	21|14
Males	12	10|6	13	9|7
Females	14	10|6	15	12|7
Litters	12		12	
Litter size (M ± SD)	5.9 ± 1.0		6.2 ± 0.6	
Litter size (range)	3–7		5–7	
Adult mice per cage (M ± SD)	3.2 ± 1.0		2.8 ± 1.0	
Adult mice per cage (range)	1–5		1–4	

Histo = histology (HE, MAP2, MBP, IBA1, GFAP); mol = molecular (fatty acids, phospholipids, Western Blot). Litter size was reported after culling at P10. ( ) = outliers omitted from all behavioral analyses due to severe repetitive turning behavior.

## Data Availability

The original contributions presented in the study are included in the article and Appendix A. Further inquiries can be directed to the corresponding author (C.T.).

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
