# Peer review of "Nutritional Supplementation Reduces Lesion Size and Neuroinflammation in a Sex-Dependent Manner in a Mouse Model of Perinatal Hypoxic-Ischemic Brain Injury"

_nutrients, 2021, doi:10.3390/nu14010176_

Round 1

Reviewer 1 Report

This could be an interesting paper dealing with a much-applied research showing a slight but significant improvement of brain lesions and inflammation markers in a mouse model of perinatal ischemic brain injury. The main concern is the mixture of micronutrients used. Looking at the results if seems that only EPA and DHA might be enough to fulfil the same result. There is no result showing a real effect of all the other components of the diet. Can the authors comment on this and better sustain the election of this nutritional supplement with this specific composition?

Standard deviation was calculated at 10% means 10% modification of the size of the lesion

Even if the sample size was calculated at 20, I see that the number of animals per group is higher for some groups, why? Is this considered ethical when the number of animals sacrificed could be lower? (Table 2)

Table 1, it is important to include the total fat contents and the saturated fat contents of both diets. This might be an important aspect that has not been taken into consideration.

Vitamin B12, Zinc and iodide were not present at all in the vitamin and mineral mixes this is surprising. Might be that the amounts in that mixture have not been considered important they should be considered.

Table 2: animals in the experimental diet are bigger, was the difference statistically significant?

“solitary housing did not affect the outcome parameters” This affirmation should be proved or maybe include the sentence data not shown or shown as supplementary.

What does it mean P114 (line 151)

The phospholipid-bound fatty acids where quantified and identified by using standards? Which standards?

In the results section Figure 1D and E; it looks like there was a statistically significant increase in body weight with the experimental diet. How do you explain this increase?

Author Response

  • The reviewer believes this manuscript could be an interesting paper dealing with a much-applied research showing a slight but significant improvement of brain lesion and inflammation markers in a mouse model of perinatal ischemic brain injury. The reviewer’s main concern is whether the improvement is due to the mixture of micronutrients used, or whether only EPA and DHA might be enough to fulfil the same result. The reviewer asks if the authors can comment on this and better sustain the election of this nutritional supplement with this specific composition.

We thank the reviewer for raising this concern. The combination of nutrients was chosen based on previous in vitro, in vivo and clinical work done elsewhere (Van Deijk et al., 2017, Wiesmann et al., 2017, Andrew et al., 2018a+b, unpublished data). In these studies, it was shown that this combination of nutrients enhanced neuronal outgrowth more than the individual components, and that these combined supplements led to clinically significant improvements in children at risk of developing cerebral palsy and/or neurodevelopmental disabilities. Furthermore, there is a significant body of literature to support the synergistic effects of omega-3 fatty acids, uridine and choline on neuronal development (e.g. Wurtman, 2010), the important contribution of zinc, choline and vitamin B12 to phospholipid synthesis (e.g. McGee et al., 2018), and the beneficial effects of all dietary components for neurodevelopment and neuroprotection after injury (as referred to in the introduction and discussion sections). We agree with the reviewer that a more extensive argumentation better sustains the election of this nutritional composition, and have therefore elaborated on this further in the introduction in lines 51-76 (newly added sections highlighted in yellow):

Phospholipids such as phosphatidylcholine (PC) and phosphatidylethanolamine (PE) are important components of neural and glial membranes, synapses, and myelin [11]. N-3 polyunsaturated fatty acids (PUFAs) such as docosahexaenoic acid (DHA, 22:6n-3) and eicosapentaenoic acid (EPA, 20:5n-3), in addition to uridine monophosphate (UMP) and choline, are required to synthesize PC through a physiological process known as the Kennedy cycle. Synthesis of these phospholipids can be synergistically increased by dietary supplementation of their precursors, leading to enhanced functional connectivity and cognitive performance [11,12]. Zinc and vitamin B12 are also involved in PC synthesis, by acting as a cofactor for the production of methionine needed for the methylation of PE in order to generate PC [13-15]. The combined supplementation of these nutrients after injury may therefore contribute to functional repair [16].

In addition to their (synergistic) effects in enhancing brain phospholipid synthesis, the n-3 fatty acids and micronutrients are known to positively influence brain development through other mechanisms. Perinatal supply of DHA and EPA has been shown to support neural and retinal development [9]. Furthermore, the neuro-regenerative and anti-inflammatory effects of n-3 fatty acids, and particularly of DHA, have been widely studied in animal models of ischemic brain injury, and have been shown to reduce apoptosis and inflammation (see [17] for a review), while depletion impairs recovery in a model of traumatic brain injury [18]. Zinc, iodide and vitamin B12 are essential contributors to brain development through their role in neurogenesis, synaptic connectivity, DNA methylation and increasing the levels of growth factors, ultimately leading to enhanced cognition [15,19-24]. Furthermore, these micronutrients can stimulate neuronal survival after injury either directly or through the reduction of oxidative stress [21-22,25-26].

This is again reaffirmed in the discussion (lines 642-647):

The aim of this study was to test the efficacy of an experimental concept diet, containing DHA, EPA, UMP, choline, iodide, zinc, and vitamin B12, in a mouse model of perinatal HIE. The listed nutrients provide building blocks for neuronal and synaptic membranes, potentially acting synergistically to support brain development by improving synaptic connectivity, reducing neuroinflammation and enhancing neuronal survival after injury [48-51].

We agree with the reviewer that the results of this study do not directly support a synergistic effect of all nutrients in addition to the documented effects of its separate components such as the omega-3 fatty acids, because this was not assessed directly in this experimental set-up. Therefore we have been careful in applying the term “synergistic” throughout the manuscript and particularly when interpreting our results in line 634 of the discussion. We sincerely hope the reviewer agrees with this approach. Moreover, we have made this limitation more explicit in section 4.2, lines 737-744 (newly added text highlighted in yellow):

Although not directly addressed in this study, involvement of other pathways through which the experimental diet reduced lesion size after HI injury cannot be ruled out. For example, n-3 fatty acids, zinc and iodine have been known to reduce apoptosis, either directly through stimulating neuronal survival or through reduction of ROS [21-22,25-26]. Furthermore, this study was not designed to investigate effects of individual components, nor to identify potential synergy between combined dietary components. Therefore, the extent to which each of the individual dietary components contributed to the effects reported here remains subject to further investigation.

  • The reviewer states that the sample size was calculated at 20 and asks why the number of animals per group was higher for some groups. The reviewer questions whether this is considered ethical when the number of animals sacrificed could be lower (Table 2)?

We agree with the reviewer that the number of experimental animals used should be reduced to the minimum needed to answer the research question. The sample size was calculated using an alpha of 1.25%, a power of 90%, effect size (relevant alteration of the lesion by the diet) of 1.3 and a standard deviation (expected variance within the experimental groups) of 10%, leading to an n = 20 for our primary outcome (lesion size analyzed by histology, lines 105-111). The following amount of animals were included in the primary outcome of this study: control diet: HI (n=20), sham (n=20), experimental diet: HI (n=20), sham (n=21).

After re-examining the animal numbers in Table 2, we found that a small error had been made. In the control diet group, 2 HI-injured males in the first cohort were initially planned for histology but reached the humane endpoint before the end of the experiment. They were therefore not included in any of the analyses. The total group size for this group was thus n = 20 instead of n = 22, which has been corrected in Table 2 of the revised manuscript and is in accordance with the calculated requirement of n = 20 mice. We thank the reviewer for this sharp observation, and we apologize for this inconvenience.

The n = 21 in the experimental diet, sham-group for histology was due to the experiment being performed in six cohorts due to restricted capacity for behavioral tests. The first cohorts were planned for histology and the latter were planned for molecular outcome parameters. The first five cohorts were used for behavioural analyses. One mouse was transferred from molecular outcome to histological outcome to better balance the distribution of sexes for our primary outcome measure (histology). Therefore, the allocation of this mouse did not lead to the use of more animals than would have been used otherwise, nor did it lead to more discomfort for this individual, confirming that there were no ethical consequences to the addition of one extra animal to this group. The experiment was carried out in accordance with institutional regulations and national and European guidelines, and after approval from the Animal Welfare Body (Utrecht, the Netherlands).

After rerunning the analysis with the omission of n = 1 female from the sham-operated experimental diet group, resulting in a total n = 20 for all the experimental groups, the statistical analysis of our primary outcome measure (MAP2 lesion size) is unaltered from what is reported in the manuscript (main effect of HI, p < .001, interaction effect HI x diet, p < .05, interaction effect HI x sex, p < .10, interaction effect HI x diet x sex, p  < .05). We are therefore confident that the inclusion of n = 1 for the sake of equal sex distribution has not increased our risk of Type 1 error (false positive).

  • The reviewer states that in Table 1, it is important to include the total fat contents and the saturated fat contents of both diets. The reviewer thinks this might be an important aspect that has not been taken into consideration.

We agree with the reviewer that this is an important consideration. The total amount of fats was kept consistent between diets, as can be found in Table 1 (7g/100g for both diets). The total saturated fatty acid content was also not different between the diets (26,55% in the control diet vs. 26,10% in the experimental diet). In agreement with the reviewer’s suggestion, the saturated fatty acid content has been added at the bottom of Table 1.

  • The reviewer states that vitamin B12, zinc and iodide were not present at all in the vitamin and mineral mixes. The reviewer asks whether the amounts in that mixture have not been considered important, while they should be considered.

We appreciate the reviewer pointing this out, and agree that vitamin B12, zinc and iodide are pivotal to a healthy diet. In our study, vitamin B12, zinc and iodide were included in the vitamin and mineral mix that is present in the commonly used AIN93G diet. Therefore, the control the diet contains standard amounts of these supplements. The experimental diet has been supplemented with these nutrients on top of the standard amounts present in the control diet. We agree with the reviewer that this was stated unclearly in Table 1 and has been clarified in the current manuscript.

  • The reviewer asks whether the animals in the experimental diet in Table 2 are bigger, and whether this difference was statistically significant?

We thank the reviewer for their question. If the reviewer refers to the weight of the animals, it is indeed reported that the experimental diet mice were heavier on average than the control diet mice (main effect of diet, p < .01, section 3.1). If the reviewer refers to the litter size mentioned in Table 2, this difference was not statistically significant between diets (p = .38).

  • The reviewer cites that “solitary housing did not affect the outcome parameters”. The reviewer asks whether this affirmation is proved or a sentence data not shown or shown as supplementary can be included.

We agree with the reviewer that it is important to take into account the potential confounding effects of solitary housing on behavior and other outcome measures. We have therefore statistically checked the scores of these three mice within their respective experimental group and have ascertained that they fell within the group range and were not detected as statistically significant outliers. Moreover, in preliminary analyses, the statistical models were run with the factor “cage” as a random predictor, which did not contribute to explaining any of the variance for our outcome measures and was thus omitted as a factor from the final analyses. As per the reviewer’s suggestion, we have added “data not shown” to this section in line 152 for sake of conciseness and comprehensibility.

  • The reviewer asks what P114 means (line 151).

We thank the reviewer for pointing out the inconsistent use of this abbreviation. Throughout the manuscript, “P” stands for postnatal day. This abbreviation has been added to lines 132 and has been adjusted in the discussion section where it had been written in full in lines 784 and 807.

  • The reviewer asks whether the phospholipid-bound fatty acids where quantified and identified by using standards, and which standards were used.

We agree that the use of standards is an important consideration in the quantification of phospholipid-bound fatty acids. The internal standard that was used was a known amount of C19:0 PC, as mentioned in the Material & Methods section 2.7: Phospholipid-bound fatty acids as follows:

A known amount of C19:0 PC was added as an internal standard to 400 µl sample (1:50 brain homogenate) and subsequently extracted according to Bligh & Dyer [45].

  • The reviewer refers to the results section Figure 1D and E; where a statistically significant increase in body weight with the experimental diet was shown. The reviewer asks how this increase can be explained.

We thank the reviewer for their thought-provoking question and have elaborated on this in section 4.3 of the discussion, lines 781-794 (see below, added text highlighted in yellow), where we hypothesize that the experimental diet led to a more favorable growth trajectory, supporting better brain development. However, measures of functional weight gain were not taken in this study and therefore direct conclusions cannot be drawn based on our data.

Weight gain is an important indicator of growth during the neonatal period, as it signifies better overall growth and head circumference, in turn enabling brain development [8]. Although pups of both diet groups had the same weight around birth, those that were subjected to HI injury on P9 had a slower growth trajectory, which may indicate impaired development. Altered growth patterns have also been reported in children suffering from cerebral palsy [67-68]. In the current study, HI-injured mice that received the experimental diet gained more weight than HI-injured control diet mice, which may indicate a more favorable growth trajectory. In a clinical setting, n-3 supplementation has been shown to increase weight at discharge and head circumference z-scores in very low birth weight infants ( < 1500g, [69]). Together, these results may indicate that the experimental diet used in the current study supports functional weight gain, although future studies should assess body length or lean body mass to give a more accurate indication of functional weight gain.

Reviewer 2 Report

Very difficult and very interesting experimental work that may influence the nutritional practice of newborns with hypoxic-ischemic brain injury.

Author Response

  • The reviewer states that this is very difficult and very interesting experimental work that may influence the nutritional practice of newborns with hypoxic-ischemic brain injury.

We thank the reviewer for reviewing our manuscript, and appreciate their view on its potential implications.

Round 2

Reviewer 1 Report

I think that the manuscript has been sufficiently improved to warrant publication in Nutrients as it is. I thank you the authors for all the responses and explanations.